**Data Availability Statement:** The PPMI dataset can be obtained from the study website (www.ppmi-info.org) upon application. The NeuroX

# Meta-analysis of whole-exome sequencing data from two independent cohorts finds no evidence for rare variant enrichment in Parkinson disease associated loci

Johannes Jernqvist Gaare[1,2], Gonzalo Nido[1,2], Christian Dölle[1,2], Paweł Sztromwasser[3,4,5], Guido Alves[6,7], Ole-Bjørn Tysnes[1,2], Kristoffer Haugarvoll[1,2], Charalampos Tzoulis[1,2]*

1 Department of Neurology, Haukeland University Hospital, Bergen, Norway, 2 Department of Clinical Medicine, University of Bergen, Bergen, Norway, 3 Department of Clinical Science, University of Bergen, Bergen, Norway, 4 Computational Biology Unit, Institute of Informatics, University of Bergen, Bergen, Norway, 5 Department of Biostatistics and Translational Medicine, Medical University of Lodz, Lodz, Poland, 6 The Norwegian Centre for Movement Disorders and Department of Neurology, Stavanger University Hospital, Stavanger, Norway, 7 Department of Chemistry, Bioscience and Environmental Engineering, University of Stavanger, Stavanger, Norway

* charalampos.tzoulis@nevro.uib.no, charalampos.tzoulis@helse-bergen.no

## Abstract

Parkinson disease (PD) is a complex neurodegenerative disorder influenced by both environmental and genetic factors. While genome wide association studies have identified several susceptibility loci, many causal variants and genes underlying these associations remain undetermined. Identifying these is essential in order to gain mechanistic insight and identify biological pathways that may be targeted therapeutically. We hypothesized that gene-based enrichment of rare mutations is likely to be found within susceptibility loci for PD and may help identify causal genes. Whole-exome sequencing data from two independent cohorts were analyzed in tandem and by meta-analysis and a third cohort genotyped using the NeuroX-array was used for replication analysis. We employed collapsing methods (burden and the sequence kernel association test) to detect gene-based enrichment of rare, protein-altering variation within established PD susceptibility loci. Our analyses showed trends for three genes (*GALC*, *PARP9* and *SEC23IP*), but none of these survived multiple testing correction. Our findings provide no evidence of rare mutation enrichment in genes within PD-associated loci, in our datasets. While not excluding that rare mutations in these genes may influence the risk of idiopathic PD, our results suggest that, if such effects exist, much larger sequencing datasets will be required for their detection.

## Introduction

Parkinson disease (PD) is a complex disorder influenced by the crosstalk between genetic and environmental factors [1]. Monogenic causes account for a small fraction of cases, whereas the

dataset can be obtained through dbGaP (dbGaP Study Accession: phs000918.v1.p1). The ParkWest dataset is currently not publicly available due to limitations set by the regional ethical board approval and study consent form, but will be made available from the Neuromics Lab upon request (https://neuromics.org/contact/).

**Funding:** CT, Grant number 240369/F20 from the Norwegian Research Council (www.nfr.no). The funders had no role in study design, data collection and analysis, decision to publish, or preparation of the manuscript. CT, grant number 911903 from the Regional Health Authority of Western Norway (www.helse-vest.no). The funders had no role in study design, data collection and analysis, decision to publish, or preparation of the manuscript. JJG, grant number 911988 from the Regional Health Authority of Western Norway (www.helse-vest.no). The funders had no role in study design, data collection and analysis, decision to publish, or preparation of the manuscript.

**Competing interests:** The authors have declared that no competing interests exist.

vast majority of patients have idiopathic disease. While genome-wide association studies (GWAS) have revealed several susceptibility *loci* for idiopathic PD, these collectively explain only a fraction of the disorder's estimated heritability, and most have not been linked to pathways which can be targeted by therapies [2]. This is partly due to the uncertainty regarding which genes actually drive the GWAS signals.

The associated variants in GWAS are typically located in noncoding regions of the genome and assumed to be in linkage disequilibrium (LD) with causative variants in nearby genes [3]. Methods to identify candidate genes from GWAS range from simply choosing the closest gene to more sophisticated algorithms [4], but all are, in essence, inferential by nature. Next generation sequencing technologies have enabled us to investigate the impact of rare genetic variation, which is theorized to explain parts of the "missing heritability" in complex diseases [5]. In PD, rare variants have been implicated in sporadic disease both at the gene- [6] and pathway level [7,8]. Whether rare variants can explain GWAS signals in PD, remains, however, unknown.

We hypothesized that gene-based enrichment of rare, protein-altering variation is likely to be found in regions tagged by single nucleotide polymorphisms (SNPs) associated with PD in GWAS, and may help identify the causal genes driving these associations. To test our hypothesis, we selected genes with variants in LD with associated SNPs from the most recent GWAS meta-analysis[9], and tested for enrichment of rare, protein-altering variants in whole-exome sequencing data from two independent cohorts.

## Methods

### Cohorts and sequencing

The Norwegian whole-exome sequencing (WES) cohort comprised 191 patients with PD from the Norwegian ParkWest study [10] and 219 controls. The control group consisted of individuals with testis cancer (n = 167) and acoustic neuroma (n = 52) who had been recruited and examined at our hospital and had no clinical signs of neurodegenerative- or other neurological disorders. DNA was extracted from blood by routine procedures and sequenced at HudsonAlpha Institute for Biotechnology (Huntsville, Alabama) on the Illumina HiSeq platform using paired-end 100 bp sequencing and Roche-NimbleGen Sequence Capture EZ Exome v2 (173 controls) and v3 (all PD and 46 controls) capture kits. Reads were mapped to the hg19 (GRCh37) reference genome using BWA v0.6.2 [11], PCR duplicates removed with Picard v1.118 [12], and the alignment refined using Genome Analysis Toolkit (GATK) v3.3.0 [13] applying base quality score recalibration and realignment around indels recommended in the GATK Best Practices workflow [14,15]. Variants were called in all samples using GATK HaplotypeCaller [13] with default parameters. Next, Variant Quality Score Recalibration (VQSR) was performed using 99.9% sensitivity threshold [13]. The remaining variants were filtered against the intersection of capture targets (v2 and v3) using BEDtools [16] and VCFtools [17]. Variants with total depth below 10X were marked as unknown genotype (no-call) using BCFtools [18]. In addition, we used a cutoff of at least 6 reads supporting each variant (alternated allele). Indels were removed prior to downstream analyses. The depth distribution for all variants and variants of interest is shown in S1 Fig.

Additional whole-exomi sequencing data was obtained from the Parkinson Progression Markers Initiative (PPMI) [19]. WES data was available from 640 individuals (459 cases and 181 controls). Control subjects were individuals without PD 30 years or older, without first degree relatives with PD. Sequencing was performed on the Illumina HiSeq 2500 platform using the Illumina Nextera Rapid Capture Expanded Exome Kit and paired-end 100 bp reads.

Calling and alignment were performed by the PPMI. Indels were removed prior to variant quality control using VCFTools [17].

SNP-chip data was obtained from the International Parkinson's Disease Genomics Consortium (IPDGC) (dbGaP Study Accession: phs000918.v1.p1). The dataset consisted of 11,402 individuals (5,540 cases and 5,862 controls) genotyped on the NeuroX array, comprising approximately 240,000 standard Illumina exome variants and 24,000 custom variants focusing on neurological diseases [20,21].

## Individual and variant quality control

Sequencing and genotype data were recoded into binary PLINK input format, and quality control of individual and SNP data was performed for all three cohorts separately. Individuals were excluded if they had an individual genotype missingness rate of $> 2\%$, heterozygosity outside +/- 3 standard deviations (calculated for common and rare variants separately), cryptic relatedness (IBD $> 0.2$), conflicting sex assignment or non-European ancestry. Population stratification was studies using multi-dimensional scaling against the HapMap populations [22]. Variants were removed if they had a genotyping rate $< 98\%$, different call rates in cases and controls ($p > 0.02$) or departure from the Hardy-Weinberg equilibrium ($p < 10^{-5}$). Only autosomes were kept for downstream analyses. Principal component analysis was performed using Eigensoft [23,24] with standard filtering settings. ANOVA of the 10 first principal components was performed with the significance level set to $p < 0.01$. Significant principal components were included as covariates in all downstream analyses. Outside of the principal component analysis, all quality control procedures were performed using PLINK v1.90 [25] and R [26].

## Annotation and subset filtering

The datasets were annotated using ANNOVAR [27] according to the RefSeq gene transcripts, and variants classified as nonsynonymous, stop-gain, stop-loss or splicing were extracted for further analysis. Rare variants were defined as having a minor allele frequency (MAF) of $< 1\%$ in the non-Finnish European population in gnomAD [28].

## Selection of genes of interest

Genomic regions associated with PD where extracted from the largest and most recent, to date, meta-analysis of GWASes, which identified 90 SNPs associated with PD at genome-wide significance level [9]. We defined genes of interest as any gene containing a variant in LD within a 2 megabase window around any of these 90 SNPs, with the threshold of LD set to $R^2$ $> 0.5$. If a variant in LD was localized in an intergenic region, the nearest gene was included. LD calculations were available from the supplementary material of the original study [9], and a total of 303 genes fit the inclusion criteria (S1 Table).

## Genetic association analyses

For each cohort, genes of interest were analyzed by two different tests: the burden test and the sequence kernel association test (SKAT) [29], using the SKAT R package v1.3.2.1 [30] with default settings. Statistically significant principal components, as determined by an ANOVA of the first 10 principal components with significance level cutoff set to $p < 0.01$, were added as covariates to all downstream analyses. Only genes with variants in both WES cohorts (Park-West and PPMI) and at least two or more variants across cohorts were included. The meta-analysis was performed using the MetaSKAT R package v0.60 [31], using the same burden test

and SKAT as described above in a meta-analysis framework. For the meta-analysis, we hypothesized that genetic effects should be homogenous across studies, meaning that the same mutation should have the same direction of effect in both cohorts. NeuroX was used as a replication cohort for the results from the WES analyses, and analyzed using the same methods. Only variants defined as rare were included. All p-values were corrected for multiple comparisons using FDR (Benjamini-Hochberg) [32].

### Gene set analyses

In addition to the single gene analyses, enrichment of genetic variation across all genes of interest was explored in a gene set analysis. Only rare variants were included, and the combined gene set was analyzed by burden and SKAT tests using the same methods and statistical tools as for the single gene analyses. A subset of loss-of-function (LoF) variants (containing only splicing, stop-loss and stop-gain variants) was extracted and similarly analyzed.

### Ethical considerations

These studies were approved by the Regional Committee for Medical and Health Research Ethics, Western Norway (REK 131/04), and all subjects gave written, informed consent. All research was performed in accordance with the relevant guidelines and regulations.

## Results

Using the inclusion criteria outlined previously, 168 genes of interest were analyzed in the single gene analyses, comprising a total of 543 rare nonsynonymous, stop-gain, stop-loss or splicing variants in the ParkWest cohort, and 1135 in the PPMI cohort. 160 of these genes were available for replication analysis in the NeuroX dataset, comprising a total of 1380 variants. For the gene set analysis, the number of included variants was 554 in the ParkWest, 1341 in the PPMI and 1534 in the NeuroX cohorts. A total of 14 LoF variants were identified in the ParkWest cohort, 17 in the PPMI cohort and 40 in the NeuroX cohort.

Gene-based analyses indicated three genes with nominally significant p-values (uncorrected $p < 0.05$) across multiple cohorts: *GALC*, *SEC23IP* and *PARP9*. However, no gene reached statistical significance surviving multiple testing correction in either of the cohorts or the meta-analysis (see S2 Table). Similarly, there were no statistically significant results in the gene set analyses (see S3 Table). The top results of the gene enrichment analyses, ranked by nominal p-value in the meta-analysis, are shown in the Tables 1 and 2.

## Discussion

Our analyses revealed no statistically significant enrichment of rare variants in genes implicated by previous GWAS in PD. Three genes (*GALC*, *SEC23IP* and *PARP9*) showed trends across multiple cohorts, but none survived multiple testing correction. Nalls et al [9] conducted rare variant burden analysis for *SEC23IP* finding no enrichment signal. Thus, it is highly unlikely that *SEC23IP* is involved in PD. The variant tagging *PARP9* (rs55961674) is a weak expression quantitative trait loci (eQTL) for *PARP9* in some tissues (nerve and thyroid) [33]. However, it is also a strong splicing QTL (sQTL) for *KPNA1*, suggesting that this is a more likely candidate gene. Finally, the variant tagging *GALC* (rs979812) is a strong eQTL for *GALC*, supporting a potential role in PD [33]. *GALC* encodes the enzyme galactocerebrosidase, and mutations in this gene cause Krabbe disease, a lysosomal storage disorder [34]. Current evidence suggests that lysosomal dysfunction plays a key role in PD [35], and rare mutations in a broad range of genes causing lysosomal storage disorders have been associated with PD

**Table 1. Top results for burden-based gene enrichment analyses.**

| Gene | ParkWest | | | PPMI | | | Meta | | | NeuroX | | |
|---|---|---|---|---|---|---|---|---|---|---|---|---|
| | Variants | P-value | FDR | Variants | P-value | FDR | Variants | P-value | FDR | Variants | P-value | FDR |
| SEC23IP | 3 | 0.0276 | 0.9035 | 9 | 0.0332 | 0.9191 | 11 | 0.0040 | 0.6669 | 10 | 0.6819 | 0.9037 |
| PARP9 | 3 | 0.0819 | 0.9035 | 9 | 0.0730 | 0.9191 | 11 | 0.0110 | 0.7058 | 7 | 0.0908 | 0.6353 |
| GALC | 4 | 0.8111 | 0.9035 | 5 | 0.0032 | 0.5335 | 8 | 0.0210 | 0.7058 | 7 | 0.2607 | 0.8180 |
| NFKB2 | 1 | 0.0180 | 0.9035 | 7 | 0.2543 | 0.9191 | 7 | 0.0333 | 0.7058 | 6 | 0.7477 | 0.9037 |
| ATP2A1 | 2 | 0.0518 | 0.9035 | 10 | 0.2232 | 0.9191 | 12 | 0.0416 | 0.7058 | 7 | 0.6897 | 0.9037 |
| PBXIP1 | 1 | 0.3142 | 0.9035 | 8 | 0.0656 | 0.9191 | 9 | 0.0457 | 0.7058 | 10 | 0.6148 | 0.9037 |
| CASR | 1 | 0.6461 | 0.9035 | 4 | 0.0276 | 0.9191 | 4 | 0.0457 | 0.7058 | 6 | 0.5906 | 0.9037 |
| ITGA8 | 2 | 0.1473 | 0.9035 | 10 | 0.1905 | 0.9191 | 12 | 0.0533 | 0.7058 | 18 | 0.6969 | 0.9037 |
| VPS13C | 26 | 0.2533 | 0.9035 | 24 | 0.1276 | 0.9191 | 42 | 0.0557 | 0.7058 | 49 | 0.0636 | 0.5420 |
| CTSB | 3 | 0.9743 | 0.9801 | 15 | 0.0279 | 0.9191 | 17 | 0.0688 | 0.7058 | 10 | 0.4903 | 0.9037 |

Genes are ranked by p-value in the meta-analysis. The FDR-column contains p-values after applying false discovery rate-correction.

[7]. Mutations of *GBA* in particular, the gene encoding the enzyme glucosylcerebrosidase that carries out a very similar reaction to that of galactocerebrosidase, are the most common genetic risk factor for PD and this association is driven by both common [36] and rare variants [37]. A role for *GALC* in α-synucleinopathies is therefore not farfetched [38].

Taken together, our findings provide no evidence of rare mutation enrichment in PD GWAS loci, in our datasets. These results do not support our initial hypothesis that gene-based enrichment of rare mutations can be helpful in identifying causal genes in PD-associated loci. It should be stressed that these findings do not disprove the hypothesis that rare mutations in these genes may influence the risk of idiopathic PD. They do, however, suggest that if such effects exist, much larger sequencing datasets will be required for their detection.

A few studies with similar approaches to ours have previously been published, using older GWAS data. Foo et al [39] probed 39 genes implicated in PD by GWAS and described enrichment of rare missense variation in *LRRK2*. Sandor et al [40] investigated 329 genes located within GWAS loci, and detected a possible enrichment of missense variation, including both common and rare mutations in their analysis, across the complete gene set. Finally, Jansen et al [41] used a Prix fixe strategy to select one candidate gene per GWAS locus, and detected

**Table 2. Top results from SKAT-based gene enrichment analyses.**

| Gene | ParkWest | | | PPMI | | | Meta | | | NeuroX | | |
|---|---|---|---|---|---|---|---|---|---|---|---|---|
| | Variants | P-value | FDR | Variants | P-value | FDR | Variants | P-value | FDR | Variants | P-value | FDR |
| CASR | 1 | 0.6461 | 0.9008 | 4 | 0.0012 | 0.2089 | 4 | 0.0029 | 0.4900 | 6 | 0.7926 | 0.9880 |
| PARP9 | 3 | 0.1087 | 0.9008 | 9 | 0.1820 | 0.9393 | 11 | 0.0215 | 0.8873 | 7 | 0.0311 | 0.5528 |
| GALC | 4 | 0.0460 | 0.9008 | 5 | 0.0366 | 0.9393 | 8 | 0.0311 | 0.8873 | 7 | 0.8539 | 0.9880 |
| NFKB2 | 1 | 0.0180 | 0.9008 | 7 | 0.9714 | 0.9944 | 7 | 0.0381 | 0.8873 | 6 | 0.2684 | 0.9257 |
| SEC23IP | 3 | 0.1313 | 0.9008 | 9 | 0.1113 | 0.9393 | 11 | 0.0469 | 0.8873 | 10 | 0.3037 | 0.9320 |
| SCARB2 | 2 | 0.1983 | 0.9008 | 3 | 0.4048 | 0.9393 | 4 | 0.0829 | 0.8873 | 11 | 0.5043 | 0.9613 |
| BTNL2 | 1 | 0.7423 | 0.9008 | 8 | 0.1152 | 0.9393 | 9 | 0.1085 | 0.8873 | 14 | 0.3768 | 0.9494 |
| CTSB | 3 | 0.4239 | 0.9008 | 15 | 0.1788 | 0.9393 | 17 | 0.1153 | 0.8873 | 10 | 0.0664 | 0.7225 |
| PAM | 5 | 0.1454 | 0.9008 | 13 | 0.2638 | 0.9393 | 17 | 0.1188 | 0.8873 | 11 | 0.8316 | 0.9880 |
| TUFM | 1 | 0.3609 | 0.9008 | 2 | 0.5208 | 0.9393 | 2 | 0.1205 | 0.8873 | 1 | 0.1826 | 0.9257 |

Genes are ranked by p-value in the meta-analysis. The FDR-column contains p-values after applying false discovery rate-correction.

rare variation association signals in *LRRK2*, *STBD1* and *SPATA19*. While we could not replicate enrichment for any of these genes in our datasets, it should be noted that our sample size (n = 1050) is smaller than that of Jansen et al [6,41].

In addition to rare variant enrichment analyses, several other methodologies have been employed to nominate causal genes from GWAS loci. eQTL studies integrate genotype and gene expression date, to identify genes whose expression is regulated by PD associated SNPs [42–46]. The effect of non-coding genetic variation on splicing of pre-mRNA (splicing QTLs or sQTLs) has also recently been highlighted and used to further explore possible causal genes in PD [47]. Finally, epigenetic quantitative trait loci, such as DNA methylation (mQTL), have also been used in combination with GWAS and eQTL data with variable success [48].

PD is a complex disease of heterogeneous etiology. While there is a clear genetic component, as evidenced by twin studies [49], known risk loci are primarily common mutations which, collectively, only explain a fraction of the total estimated heritability [9]. As for other complex disorders, much of the unexplained heritability is believed to be caused by rare variants [50]. Multiple studies have linked common mutations, either through the use of polygenic risk scores [51] or machine learning algorithms [42], to motor progression and cognitive decline. In addition, common genetic variation has also been shown to impact drug responsiveness in PD [52]. Similar applications of rare variants could potentially increase the predictive precision of these models and provide clinicians with a powerful tool to individualize treatment and follow-up for PD patients.

In conclusion, our results indicate that rare variant enrichment alone is unlikely to be helpful in identifying causal risk genes for PD in small to moderately sized cohorts. Larger studies are needed to determine if rare variant enrichment with small effect sizes are present in these genes. Future studies will likely need to integrate multiple types of data, including GWAS, sequencing and various forms of QTL analyses as well as functional experiments in order to better characterize the effects of rare coding variation in PD and identify novel genes and biological pathways.

## Supporting information

**S1 Fig. Variant depth distribution.** A) Depth distribution of all variants called across all samples. B) Depth distribution for the subset of variants called within the predefined regions of interest across all samples. Red bars represent heterozygous variants (0/1), and blue bars represent homozygous (1/1) variants. The vertical dashed line represents the cutoff of minimum 10 reads employed in the analyses.
(PDF)

**S1 Table. Genes of interest.**
(PDF)

**S2 Table. Complete results from gene-based rare variant enrichment analyses.**
(PDF)

**S3 Table. Gene set analyses.**
(PDF)

## Acknowledgments

Data used in the preparation of this article were obtained from the Parkinson's Progression Markers Initiative (PPMI) database (www.ppmi-info.org/data). For up-to-date information on the study, visit www.ppmi-info.org.

## Author Contributions

**Conceptualization:** Charalampos Tzoulis.

**Data curation:** Paweł Sztromwasser.

**Formal analysis:** Johannes Jernqvist Gaare.

**Funding acquisition:** Charalampos Tzoulis.

**Methodology:** Johannes Jernqvist Gaare, Gonzalo Nido, Charalampos Tzoulis.

**Project administration:** Charalampos Tzoulis.

**Supervision:** Kristoffer Haugarvoll, Charalampos Tzoulis.

**Writing – original draft:** Johannes Jernqvist Gaare, Charalampos Tzoulis.

**Writing – review & editing:** Johannes Jernqvist Gaare, Gonzalo Nido, Christian Dölle, Guido Alves, Ole-Bjørn Tysnes, Kristoffer Haugarvoll, Charalampos Tzoulis.

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
