## [Decision Letter · Decision Letter 0]

29 Jul 2020

PONE-D-20-16531

No evidence for rare variant enrichment in Parkinson disease associated loci

PLOS ONE

Dear Dr. Tzoulis,

Thank you for submitting your manuscript to PLOS ONE. After careful consideration, we feel that it has merit but does not fully meet PLOS ONE’s publication criteria as it currently stands. Therefore, we invite you to submit a revised version of the manuscript that addresses the points raised during the review process.

We look forward to receiving your revised manuscript.

Kind regards,

Giuseppe Novelli

Academic Editor

PLOS ONE

Journal Requirements:

"For up-to-date information on

244 the study, visit www.ppmi-info.org. PPMI – a public-private partnership – is funded by the

245 Michael J. Fox Foundation for Parkinson’s Research and funding partners, including AbbVie,

246 Allergan, Avid Radiopharmaceuticals, Biogen, BioLegend, Bristol-Myers Squibb, Celgene,

247 Denali, GE Healthcare, Genentech, GlaxoSmithKline, Lilly, Lundbeck, Merck, Meso Scale

248 Discovery, Pfizer, Prevail Therapeutics, Piramal, Roche, Sanofi Genzyme, Servier, Takeda,

249 Teva, and UCB.".

i) We note that you have provided funding information that is not currently declared in your Funding Statement. However, funding information should not appear in the Acknowledgments section or other areas of your manuscript. We will only publish funding information present in the Funding Statement section of the online submission form.

ii) Please remove any funding-related text from the manuscript and let us know how you would like to update your Funding Statement. Currently, your Funding Statement reads as follows:

 "CT, Grant number 240369/F20 from the Norwegian Research Council (www.nfr.no). The funders had no role in study design, data collection and analysis, decision to publish, or preparation of the manuscript.

CT, grant number 911903 from the Regional Health Authority of Western Norway (www.helse-vest.no). The funders had no role in study design, data collection and analysis, decision to publish, or preparation of the manuscript.

JJG, grant number 911988 from the Regional Health Authority of Western Norway (www.helse-vest.no). The funders had no role in study design, data collection and analysis, decision to publish, or preparation of the manuscript.".

iii) Additionally, because some of your funding information pertains to [commercial funding], we ask you to provide an updated Competing Interests statement, declaring all sources of commercial funding. 

iv) In your Competing Interests statement, please confirm that your commercial funding does not alter your adherence to PLOS ONE Editorial policies and criteria by including the following statement: "This does not alter our adherence to PLOS ONE policies on sharing data and materials.” as detailed online in our guide for authors  http://journals.plos.org/plosone/s/competing-interests.  If this statement is not true and your adherence to PLOS policies on sharing data and materials is altered, please explain how.

v) Please include the updated Competing Interests Statement and Funding Statement in your cover letter. We will change the online submission form on your behalf.

Reviewers' comments:

Reviewer's Responses to Questions

**Comments to the Author**

1. Is the manuscript technically sound, and do the data support the conclusions?

Reviewer #1: Partly

Reviewer #2: Yes

2. Has the statistical analysis been performed appropriately and rigorously? 

Reviewer #1: Yes

Reviewer #2: Yes

3. Have the authors made all data underlying the findings in their manuscript fully available?

Reviewer #1: Yes

Reviewer #2: Yes

4. Is the manuscript presented in an intelligible fashion and written in standard English?

Reviewer #1: Yes

Reviewer #2: Yes

5. Review Comments to the Author

Reviewer #1: The manuscript describes a sequencing-based association study for rare variants identification in three different Parkinson’s disease (PD) cohorts. Despite the interesting methodological approach, the study failed to replicate previous results or identify rare variants associated with PD phenotype after multiple testing corrections. The authors performed the genetic association analyses using the burden and sequence kernel association test (SKAT) with default settings, considering only homogeneous genetic effects. However, the authors should explain in detail why they did to not use heterogeneous collapsing method such as Variance-component tests or adaptive methods like the adaptive burden tests (PMID: 30214655; PMID: 24995866), which may increase the detection power taking into account heterogeneous genetic effects, which should not be neglected considering the complex heterogeneity of PD. In my opinion, heterogeneous collapsing methods could really improve this work, providing positive results or confirming the negative outcome. Moreover, the discussion should be extended providing more information concerning the complex heterogeneity of PD in terms of genetic and non-genetic factors, in order to draw a broader context in which the identification of rare variants could be useful to explain parts of the missing heritability of PD and to be implemented into the clinical practice in the perspective of a precision medicine approach for the treatment and management of PD patients (PMID: 31521533; PMID: 30190701; PMID: 30556112).

Reviewer #2: This article incorporates both a cohort-study and a meta-analysis. The authors made an initial assumption: gene-based enrichment of rare mutation is likely to be found in regions tagged by SNPs associated with PD in GWAS. They conducted a WES analysis on two different cohort including patients with PD and controls, one from the Norwegian Park West study (191 with PD; 219 controls) and the other from the Parkinson Progression Markers Initiative or PPMI (459 with PD; 181 controls) and they used as replication cohort for their results a SNP-chip data obtained from the International Parkinson’s Disease Genomic Consortium (IPDGC), genotyped on the Neuro X array. The genes of interest (total 303) were selected on the base of the 90 SNPs associated with PD identified by Nalls MA et al, in their GWAS study. The genetic association analyses were conducted considering only those genes which presented two or more variants and with variants in both Park West and PPMI cohorts. In addition, they started a gene set analysis using the same methods. Only variants classified as rare (MAF < 1%) were included. In the end, despite three genes (GALC, SEC23IP, and PARP9) showed trends across multiple cohorts, none of them survived multiple testing correction, revealing no statistically significant enrichment of rare variants in genes implicated by previous GWAS in PD. Nonetheless the authors point out that “these findings do not disprove the hypothesis that rare mutations in these genes may influence the risk of idiopathic PD” also in sight of the cohort size.

To my concern, the major flaws of the study are:

• Size cohort: The size cohort proposed, as pointed out by the authors themselves, is pretty small, considering the fact that PD has a multifactorial etiology and that a genetic cause can be identified only in a small number of patients.

• Selection of patients and controls: The selection modality for control patients of the PPMI study wasn’t specified. Moreover, the analysis wasn’t stratified and standardized for age and sex, considering also the important difference between Late Onset PD (LOPD) and Young Onset PD (YOPD).

Minor flaws:

• Since the authors performed also a meta-analysis it should be indicated in the title.

• The latest version (04/03/2019) of the GWAS study on which was based the gene selection (Nalls MA et al; n. 9 in the bibliography) is actually available as a preprint with the title: “Expanding Parkinson’s disease genetics: novel risk loci, genomic context, causal insights and heritable risk” and not as “Parkinson’s disease genetics: identifying novel risk loci, providing causal insights and improving estimates of heritable risk”. Thus, I would advise the authors to double-check the bibliography.

• Text errors: 1) line 108: (5,540 cases and 5,862 controls?); 2) these findings.

6. PLOS authors have the option to publish the peer review history of their article (what does this mean?). If published, this will include your full peer review and any attached files.

Reviewer #1: No

Reviewer #2: No

---

## [Author Response · Author response to Decision Letter 0]

5 Sep 2020

Dear Dr. Giuseppe Novelli,

Academic Editor of PLOS ONE

Re: PONE-D-20-16531

Thank you for allowing us to submit a revised version of our manuscript. We wish to thank the reviewers for their thorough evaluation. Please see our point-by-point response to their comments below.

Reviewer #1:

The manuscript describes a sequencing-based association study for rare variants identification in three different Parkinson’s disease (PD) cohorts. Despite the interesting methodological approach, the study failed to replicate previous results or identify rare variants associated with PD phenotype after multiple testing corrections. The authors performed the genetic association analyses using the burden and sequence kernel association test (SKAT) with default settings, considering only homogeneous genetic effects. However, the authors should explain in detail why they did to not use heterogeneous collapsing method such as Variance-component tests or adaptive methods like the adaptive burden tests (PMID: 30214655; PMID: 24995866), which may increase the detection power taking into account heterogeneous genetic effects, which should not be neglected considering the complex heterogeneity of PD. In my opinion, heterogeneous collapsing methods could really improve this work, providing positive results or confirming the negative outcome.

We thank the reviewer for this comment. We completely agree that heterogeneous collapsing methods are more powerful than regular burden tests and can be more informative in this type of analyses. We did, in fact, employ variance component tests in our analyses taking heterogeneous genetic effects into account. We do, however, realize that our methods description may have been unclear and imprecise on this matter. We have modified this section to improve clarity (lines 150-163) and explain this in more detail below.

In our analyses, we used two tests: one was a regular burden test that does not accommodate heterogeneous effects and the other was the sequence kernel association test (SKAT). SKAT is a variance component test that allows for heterogeneous effects of variants (e.g. opposite effects on the same gene). We have modified the wording in our methods section to make it clear that different tests were employed.

In addition, we noticed that the following sentence (line 158) regarding our meta-analysis was ambiguous and confusing:

“We hypothesized that genetic effects should be homogenous across studies[…]”.

To clarify, we used the SKAT test in a meta-analysis framework, allowing for heterogeneous effects. However, we operated under the assumption that the same mutation should not have different directions of effect in our two datasets. So, if a variant was protective in the ParkWest dataset, it would not be plausible that it should be associated with increased risk in PPMI. There is a version of the meta-SKAT test that allows for different directions of effects across cohorts, but we elected to use the more conservative version of the test. However, in the single-cohort analyses, the SKAT test allows for heterogeneous effects. We have revised this part of the methods section to clarify this. An adaptive burden test could have been used as an alternative to SKAT, but they have generally been shown to be less powerful than variance component tests[1].

Moreover, the discussion should be extended providing more information concerning the complex heterogeneity of PD in terms of genetic and non-genetic factors, in order to draw a broader context in which the identification of rare variants could be useful to explain parts of the missing heritability of PD and to be implemented into the clinical practice in the perspective of a precision medicine approach for the treatment and management of PD patients (PMID: 31521533; PMID: 30190701; PMID: 30556112).

This is indeed an important perspective, and we have added a paragraph to the discussion addressing this and including the suggested references.

Reviewer #2:

This article incorporates both a cohort-study and a meta-analysis. The authors made an initial assumption: gene-based enrichment of rare mutation is likely to be found in regions tagged by SNPs associated with PD in GWAS. They conducted a WES analysis on two different cohort including patients with PD and controls, one from the Norwegian Park West study (191 with PD; 219 controls) and the other from the Parkinson Progression Markers Initiative or PPMI (459 with PD; 181 controls) and they used as replication cohort for their results a SNP-chip data obtained from the International Parkinson’s Disease Genomic Consortium (IPDGC), genotyped on the Neuro X array. The genes of interest (total 303) were selected on the base of the 90 SNPs associated with PD identified by Nalls MA et al, in their GWAS study. The genetic association analyses were conducted considering only those genes which presented two or more variants and with variants in both Park West and PPMI cohorts. In addition, they started a gene set analysis using the same methods. Only variants classified as rare (MAF < 1%) were included. In the end, despite three genes (GALC, SEC23IP, and PARP9) showed trends across multiple cohorts, none of them survived multiple testing correction, revealing no statistically significant enrichment of rare variants in genes implicated by previous GWAS in PD. Nonetheless the authors point out that “these findings do not disprove the hypothesis that rare mutations in these genes may influence the risk of idiopathic PD” also in sight of the cohort size.

To my concern, the major flaws of the study are:

• Size cohort: The size cohort proposed, as pointed out by the authors themselves, is pretty small, considering the fact that PD has a multifactorial etiology and that a genetic cause can be identified only in a small number of patients.

We thank the reviewer for this comment. We do agree that sample size is a limitation and have pointed this out in our discussion. Our study was not designed to identify single causal alleles, but to determine whether disruptive rare mutations were enriched in PD-associated genes. We have previously employed similar approaches to successfully identify pathway-enrichment in the same material. Moreover, previous studies assessing PD-associated loci were able to detect some enrichment in both single genes [2] and gene-sets [3], in spite of smaller sample sizes than our combined cohorts. Since our sample size is limited, we acknowledge that our study cannot completely rule out rare variant enrichment events in the assessed genes. However, our results indicate that such effects, if present, are likely to be small/weak, as they were not detectable in our moderately sized data set. We have expanded our concluding remarks to better acknowledge this.

• Selection of patients and controls: The selection modality for control patients of the PPMI study wasn’t specified. Moreover, the analysis wasn’t stratified and standardized for age and sex, considering also the important difference between Late Onset PD (LOPD) and Young Onset PD (YOPD).

The PPMI included control subjects above the age of 30 years without PD, and without a first degree relative with PD. We have updated the methods section to include this information. As the reviewer points out, stratifying our analysis (for age and sex) would have been interesting as previous studies have shown that PD-associated SNPs are more prevalent in early onset cases[4]. However, because of our small cohort size and the hypothesized low effect size of rare variants, a stratified analysis was not possible.

Minor flaws:

• Since the authors performed also a meta-analysis it should be indicated in the title.

Per the reviewer’s recommendation, we have changed the title accordingly to “Meta-analysis of whole-exome sequencing data from two independent cohorts finds no evidence for rare variant enrichment in Parkinson disease associated loci”.

• The latest version (04/03/2019) of the GWAS study on which was based the gene selection (Nalls MA et al; n. 9 in the bibliography) is actually available as a preprint with the title: “Expanding Parkinson’s disease genetics: novel risk loci, genomic context, causal insights and heritable risk” and not as “Parkinson’s disease genetics: identifying novel risk loci, providing causal insights and improving estimates of heritable risk”. Thus, I would advise the authors to double-check the bibliography.

We have corrected the bibliography.

• Text errors: 1) line 108: (5,540 cases and 5,862 controls?); 2) these findings.

We have corrected these errors and thank the reviewer for pointing them out.

Yours sincerely,

Charalampos Tzoulis, MD, PhD

Professor of Neurology and Neurodegeneration

Vice Director, Neuro-SysMed Center of Excellence

Department of Neurology, Haukeland University Hospital

Department of Clinical Medicine, University of Bergen

http://www.neuromics.org/

tzoulis.charalampos@helse-bergen.no

charalampos.tzoulis@nevro.uib.no

Tel: +4755975045

Fax: +4755975164

References

1. Basu S, Pan W. Comparison of statistical tests for disease association with rare variants. Genetic epidemiology. 2011;35(7):606-19. Epub 2011/07/18. doi: 10.1002/gepi.20609. PubMed PMID: 21769936.

2. Foo JN, Tan LC, Liany H, Koh TH, Irwan ID, Ng YY, et al. Analysis of non-synonymous-coding variants of Parkinson's disease-related pathogenic and susceptibility genes in East Asian populations. Human Molecular Genetics. 2014;23(14):3891-7. doi: 10.1093/hmg/ddu086.

3. Sandor C, Honti F, Haerty W, Szewczyk-Krolikowski K, Tomlinson P, Evetts S, et al. Whole-exome sequencing of 228 patients with sporadic Parkinson's disease. Sci Rep. 2017;7:41188. doi: 10.1038/srep41188. PubMed PMID: 28117402; PubMed Central PMCID: PMCPMC5259721.

4. Escott-Price V, for the International Parkinson's Disease Genomics C, Nalls MA, Morris HR, Lubbe S, Brice A, et al. Polygenic risk of Parkinson disease is correlated with disease age at onset. Annals of Neurology. 2015;77(4):582-91. doi: 10.1002/ana.24335.

---

## [Editor Report · Decision Letter 1]

15 Sep 2020

Meta-analysis of whole-exome sequencing data from two independent cohorts finds no evidence for rare variant enrichment in Parkinson disease associated loci

PONE-D-20-16531R1

Dear Dr. Tzoulis,

We’re pleased to inform you that your manuscript has been judged scientifically suitable for publication and will be formally accepted for publication once it meets all outstanding technical requirements.

Kind regards,

Giuseppe Novelli

Academic Editor

PLOS ONE
---

## [Editor Report · Acceptance letter]

17 Sep 2020

PONE-D-20-16531R1

Meta-analysis of whole-exome sequencing data from two independent cohorts finds no evidence for rare variant enrichment in Parkinson disease associated loci

Dear Dr. Tzoulis:

I'm pleased to inform you that your manuscript has been deemed suitable for publication in PLOS ONE. Congratulations! Your manuscript is now with our production department.

Kind regards,

on behalf of

Prof. Giuseppe Novelli 

Academic Editor

PLOS ONE